# Long-Term Prognostic Value in Nuclear Cardiology: Expert Scoring Combined with Automated Measurements vs. Angiographic Score

**DOI:** 10.3390/jimaging12010006

**Published:** 2025-12-25

**Authors:** George Angelidis, Stavroula Giannakou, Varvara Valotassiou, Emmanouil Panagiotidis, Ioannis Tsougos, Chara Tzavara, Dimitrios Psimadas, Evdoxia Theodorou, Charalampos Ziangas, John Skoularigis, Filippos Triposkiadis, Panagiotis Georgoulias

**Affiliations:** 1Nuclear Medicine Laboratory, University Hospital of Larissa, University of Thessaly, 41110 Larissa, Greece; giannstav@yahoo.gr (S.G.); valotasiou@med.uth.gr (V.V.); htzavara@med.uoa.gr (C.T.); pgeorgoul@med.uth.gr (P.G.); 2Medical Physics Laboratory, University Hospital of Larissa, University of Thessaly, 41110 Larissa, Greece; 3Department of Cardiology, University Hospital of Larissa, University of Thessaly, 41110 Larissa, Greece

**Keywords:** angiographic score, MPI, prognosis, software

## Abstract

The evaluation of myocardial perfusion imaging (MPI) studies is based on the visual interpretation of the reconstructed images, while the measurements obtained through software packages may contribute to the investigation, mainly in cases of ambiguous scintigraphic findings. We aimed to investigate the long-term prognostic value of expert reading of Summed Stress Score (SSS), Summed Rest Score (SRS), and Summed Difference Score (SDS), combined with the automated measurements of these parameters, in comparison to the prognostic ability of the angiographic score for soft and hard cardiac events. The study was conducted at the Nuclear Medicine Laboratory of the University of Thessaly, in Larissa, Greece. Overall, 378 consecutive patients with known or suspected coronary artery disease (CAD) were enrolled. Automated measurements of SSS, SRS, and SDS were obtained using the Emory Cardiac Toolbox, Myovation, and Quantitative Perfusion SPECT software packages. Coronary angiographies were scored according to a four-point scoring system (angiographic score). Follow-up data were recorded after phone contact, as well as through review of hospital records. All participants were followed up for at least 36 months. Soft and hard cardiac events were recorded in 31.7% and 11.6% of the sample, respectively, while any cardiac event was recorded in 36.5%. For hard cardiac events, the prognostic value of expert scoring, combined with the prognostic value of the automated measurements, was significantly greater compared to the prognostic ability of the angiographic score (*p* < 0.001). As far as any cardiac event, the prognostic value of expert scoring, combined with the prognostic value of the automated analyses, was significantly greater compared to the prognostic ability of the angiographic score (*p* < 0.001). According to our results, in patients with known or suspected CAD, the combination of expert reading and automated measurements of SSS, SRS, and SDS shows a superior prognostic ability in comparison to the angiographic score.

## 1. Introduction

Myocardial perfusion imaging (MPI) has a widely accepted role in the diagnostic investigation and therapeutic management of patients with known or suspected coronary artery disease (CAD). Additionally, MPI provides incremental value for risk assessment in these patients [1]. Previous studies have shown that the risk of myocardial infarction (MI) and cardiac death is linearly related to the extent and severity of perfusion abnormalities [2,3]. MPI was reported to have incremental prognostic value, even after stratifying for pre-imaging data. In CAD patients, normal or low-risk findings on MPI studies are linked to annual major adverse cardiovascular event rates, which are comparable to the corresponding rates in the general population [4,5]. Interestingly, the annual risk of cardiovascular events is low even in patients with angiographically documented CAD and a normal or low-risk MPI study [5].

Despite the fact that extensive reading experience is required for the accurate interpretation of MPI studies, visual assessment of the reconstructed images remains the main method of evaluation. Automated quantitation methods, using several software packages, are also available and have been incorporated into clinical practice. The Summed Stress Score (SSS), Summed Rest Score (SRS), and Summed Difference Score (SDS) are key metrics in the scintigraphic evaluation of myocardial perfusion. These parameters assess the presence of CAD by providing information on myocardial blood flow. SSS indicates perfusion defects during ischemia, SRS is linked to fixed defects, and SDS (SSS minus SRS) reveals the ischemic burden, helping determine risk. However, the major drawback of the automated MPI analyses is the inability of software packages to distinguish between actual perfusion abnormalities and artifacts [6]. Investigating the incremental diagnostic performance of automated analyses, we reported that visually defined Summed Stress Score (SSS), Summed Rest Score (SRS), and Summed Difference Score (SDS) were more strongly correlated to the angiographic data, indicating a better performance of expert scoring when compared to automated measurements [6]. Moreover, we demonstrated the superior prognostic value of expert scoring of SSS, SRS, and SDS, compared to the prognostic significance of the corresponding automated measurements [7].

In the present study, we aimed to investigate the role of expert reading of SSS, SRS, and SDS, combined with automated measurements of these parameters, as long-term prognostic markers of cardiac events, in comparison to the prognostic value of the angiographic score. The accurate evaluation of patients with known or suspected CAD, using the combination of expert scoring and automated MPI measurements, could lead to a better prognostication of this patient group, guiding therapeutic management. Therefore, given its related risks as an interventional technique, coronary angiography could be performed mostly in patients who are candidates for coronary interventions.

## 2. Materials and Methods

### 2.1. Study Population

The present research was conducted at the Nuclear Medicine Laboratory, University Hospital of Larissa, Greece. All participants gave informed consent for their complete enrolment, according to the Hospital Ethics Committee guidelines and the ethical guidelines of the Declaration of Helsinki. The study population consisted of 378 consecutive patients, with available follow-up data, who did not meet any of the exclusion criteria. Exclusion criteria included (1) no proper withdrawal of medications that can influence performance during stress testing, MPI, and the associated parameters (i.e., b-blockers, calcium channel antagonists, nitrates); (2) percutaneous coronary intervention or coronary artery bypass grafting ≤ 3 months after MPI; (3) history or other evidence of myocardial infarction; (4) severe congenital heart disease; (5) severe valvular disease; (6) non-ischemic cardiomyopathy; (7) pregnancy; and (8) qualitatively suboptimal scintigrams (presence of artifacts). Participants had been referred to our laboratory, between January 2014 and December 2018, for the performance of stress/rest myocardial perfusion single-photon emission computed tomography (SPECT). Moreover, all participants underwent coronary angiography prior to or after SPECT MPI (within a 3-month period). In particular, 282 patients underwent coronary angiography after MPI, and 96 patients underwent coronary angiography before MPI (median time interval of 28.3 days).

A brief structured interview was used in order to record the medical history of each participant. In particular, information regarding clinical features, medications, previous cardiac events, CAD risk factors, and cardiac or non-cardiac comorbidities was collected. Obesity was considered as a condition with a body mass index (BMI) score of ≥30.0 (BMI calculated as weight in kilograms divided by height in meters squared). Hypertension was defined as a systolic blood pressure of ≥140 mmHg at rest and/or a diastolic blood pressure of ≥90 mmHg at rest, or by the use of antihypertensive agents. Previous diagnosis of diabetes mellitus or lipid disorders was recorded based on the medical history data, including treatment with the corresponding medications. Finally, written instructions associated with the required radiation protection measures were given to each participant.

### 2.2. Stress Testing

The stress testing procedures were in accordance with the European Association of Nuclear Medicine (EANM) guidelines [8,9]. Overall, 202 patients underwent symptom-limited treadmill testing (Bruce protocol) after 6–12 h of fasting and avoidance of smoking or heavy intense physical activity (for at least 3 h). Data on estimated workload in metabolic equivalents (METs, using standard tables), as well as data on exercise testing and related symptoms, were collected. A total of 171 participants underwent pharmacologic testing with adenosine or regadenoson (combined with low-level exercise) due to contraindications to exercise testing or inability to achieve a satisfactory exercise level. Finally, five patients with left bundle branch block (LBBB) or an implantable pacemaker were subjected to pharmacologic stress without any form of exercise.

### 2.3. Angiographic Findings and Score

All invasive coronary angiograms (ICAs) were clinically indicated according to the medical history and the clinical characteristics of the patients and were blindly interpreted by one experienced observer. Stenosis of the vessel lumen >50% (or fractional flow reserve, FFR ≤ 0.8) was defined as hemodynamically significant, while stenoses in the left mainstem were considered equivalent to a two-vessel disease. Therefore, according to the angiographic findings, the following 4-point system (angiographic score) was used in the classification of the studies:-Normal study: 0;-One-vessel disease: 1;-Two-vessel disease: 2;-Three-vessel disease: 3.

### 2.4. SPECT MPI and Scintigraphic Findings

After the administration of technetium 99 m (99mTc) tetrofosmin (Myoview, GE Healthcare, Chicago, IL, USA), MPI studies were carried out in the supine position, using a dual-headed SPECT camera. Based on societal guidelines, injected activities were 250–400 MBq for stress and 625–1000 MBq for rest acquisitions [9]. No attenuation–scatter correction was performed.

Polar and three-dimensional mapping were carried out (GE Xeleris Software (version 3.05), Milwaukee, WI, USA), and filtered back projection was used (Butterworth Filter) for tomographic reconstruction. Two independent experienced observers blindly evaluated the acquired and reconstructed data of both stress and rest acquisitions. Radiotracer uptake was scored in each of the 17 LV segments using a 5-point scoring system (0: normal uptake; 1: mildly decreased uptake; 2: moderately decreased uptake; 3: severely decreased uptake; and 4: no uptake) [4]. This 5-point scoring system is a widely accepted tool for the quantification of stress and rest MPI images [4]. In regions with decreased counts linked to attenuation artifacts, the score was 0 [10]. SSS and SRS values were derived by adding the scores of each segment in stress and rest imaging, respectively, while SDS was calculated by subtracting SRS from SSS [11]. MPI readers were fully blinded to angiographic findings and clinical follow-up information. In 18 studies, the view of a third observer was requested, due to discordance between the two observers, and the disagreement was resolved by consensus [12].

Automated measurements of SSS, SRS, and SDS were obtained using Emory Cardiac Toolbox [ECTb (Version 3.0), Emory University, Atlanta, GA, USA], Myovation [MYO, Xeleris version 3.05, GE Healthcare, Chicago, IL, USA], and Quantitative Perfusion SPECT [QPS (Version 4.0), Cedars-Sinai Medical Center, Los Angeles, CA, USA]. An institutional normal database from 100 patients (50 males and 50 females) was created for ECTb and QPS software packages. These patients, with a low CAD likelihood and visually normal stress and rest images, had been referred to our laboratory for the performance of SPECT MPI but did not participate in the study. Therefore, automated measurements of SSS, SRS, and SDS were recorded for the participants using the institutional normal database and the ECTb and QPS software packages.

On the other hand, the MYO software package does not have a normal user database creation feature. In addition, the MYO software package does not provide standardized segmental perfusion scores. For these reasons, we converted the average segmental count values (relative to maximum pixel values in the relevant polar plot) to categorical scores according to >70%, 50–69%, 30–49%, 10–29%, and <10% thresholds, as previously described [13].

### 2.5. Follow-Up

The follow-up period of the study was at least 36 months, and the relevant data were acquired either directly, through contacting the participants via phone (or their relatives or referring physicians), or indirectly after reviewing patients’ hospital records. Hard and soft cardiac events are presented in Table 1. Cardiovascular death was defined according to the Tenth International Classification of Diseases (numbers I00–I99) as death caused by diseases of the circulatory system. In the presence of multiple causes, a reviewer, who was blinded to the hypothesis of the study, the clinical parameters, and the imaging data of the patients, evaluated the cause of death based on the death certificates.

### 2.6. Statistical Analysis

Both MI and cardiac death are binary outcomes. Quantitative variables are expressed as mean (Standard Deviation) and median (interquartile range) values, while categorical variables are expressed as absolute and relative frequencies. Receiver operating characteristic (ROC) curves were used in order to estimate the prognostic ability of the angiographic score. Afterwards, multiple logistic regression models were used in order to derive linear predictors and compare the areas under the curve. Area under the curve comparison indicates the best model for the prognosis of any cardiac event, or of hard cardiac events. In the multiple logistic regression, terms regarding experts’ score, angiography score, and software indexes were entered as independent variables in order to estimate their combined prognostic value via the models’ linear predictors. All reported *p* values are two-tailed. Statistical significance was set at *p* < 0.05, and analyses were conducted using STATA statistical software (version 13.0).

## 3. Results

Data from 378 patients (61.9% males) with a mean age of 63.8 years (SD = 9.6 years) were analyzed, and characteristics of these patients are presented in Table 2. Mean BMI was 29.5 kgr/m2 (SD = 5.4 kgr/m^2^). Symptoms were recorded in 72% of the sample, while the most frequent clinical features were angina-like symptoms (27.5%) or angina (24.9%). The median number of risk factors was three (IQR: 2–4); 74.6% of the participants suffered from hypertension, and 79.4% suffered from lipid disorders. At least one comorbidity was recorded in 22.8% of the sample; 13.2% had chronic obstructive pulmonary disease (COPD). As far as cardiac events, soft events presented in 31.7% of the sample, and hard events were seen in 11.6% of the sample, while any cardiac event was recorded in 36.5% of the participants. The mean time until any event was 67.7 months (SE = 2.69 months). The median angiographic score was 1 (IQR: 0–2).

The prognostic value of angiographic score regarding any cardiac event was significant (AUC = 0.71; 95% CI: 0.65–0.76), as well as its prognostic value regarding hard cardiac events (AUC = 0.65; 95% CI: 0.57–0.73) (Table 3). The prognostic value of expert scoring (all indexes combined) was also significant (AUC = 0.88; 95% CI: 0.84–0.91) for any cardiac event and significantly greater than the prognostic value of angiographic score (*p* < 0.001). Similarly, for hard events, the prognostic value of expert scoring (all indexes combined) was significant (AUC = 0.82; 95% CI: 0.75–0.89) and significantly greater than the prognostic value of angiographic score (*p* < 0.001).

The prognostic value of expert scoring combined with the prognostic value of the three software packages (all indexes combined) was significant (AUC = 0.91; 95% CI: 0.88–0.94) for any cardiac event. In particular, the prognostic value of the combination of expert scoring and automated analyses (all indexes combined) for any cardiac event was significantly greater compared to the prognostic performance of expert scoring (*p* < 0.001), as well as the prognostic value of angiographic score (*p* < 0.001). Furthermore, the prognostic ability of the combination of expert scoring, all automated analyses, and angiographic score was significant (AUC = 0.91; 95% CI: 0.88–0.94) (Table 4). However, the incremental predictive value of the angiographic score for ‘’any cardiac event’’ was not significant (*p* = 0.894) (Figure 1).

Similarly, the prognostic value of expert scoring combined with the prognostic value of the automated analyses of the three software packages (all indexes combined) was significant (AUC = 0.87; 95% CI: 0.81–0.92) for hard cardiac events. In particular, the prognostic value of the combination of expert scoring and automated analyses (all indexes combined) for hard cardiac events was significantly greater compared to the prognostic ability of expert scoring (*p* = 0.032), as well as the prognostic performance of angiographic score (*p* < 0.001). Moreover, the prognostic value of the combination of expert scoring, all automated analyses, and angiographic score was significant (AUC = 0.88; 95% CI: 0.82–0.93) for hard cardiac events (Table 5). Nevertheless, the angiographic score did not show a significant incremental predictive value over the combination of expert scoring and all software packages for hard cardiac events (*p* = 0.099) (Figure 2).

## 4. Discussion

Previously, we demonstrated the significantly greater prognostic value of expert scoring in comparison to automated analyses with regard to future cardiac events [7]. In the present study, we compared the prognostic performance of expert reading, combined with the corresponding automated measurements of these parameters, with the prognostic value of the angiographic score. ICA is recognized as the reference standard for the investigation of CAD, providing anatomical information, with prognostic significance, regarding the underlying coronary stenoses [14]. An accurate evaluation of the hemodynamic significance of coronary stenoses can be obtained based on the FFR measurements. However, limited information is provided by ICA for the influence of stenoses on the extent of myocardial perfusion abnormalities [15]. Our study population consisted of patients clinically referred to our laboratory in order to undergo SPECT MPI, for whom angiographic and follow-up data were available. FFR measurements were only performed in a subgroup of participants, in line with the angiographic findings and the related interventional cardiologist’s evaluation, as previously described [16,17].

According to our results, the prognostic ability of both expert reading and angiographic score was significant for any cardiac event and hard cardiac events. However, the prognostic value of the combination of expert scoring and automated MPI analyses was significantly greater compared to the prognostic performance of expert scoring (*p* < 0.001 for any cardiac event; *p* = 0.032 for hard cardiac events), as well as the prognostic ability of the angiographic score (*p* < 0.001 for any cardiac event and for hard cardiac events). Furthermore, the incremental predictive value of the angiographic score was not significant over the combination of expert scoring and automated analyses, for future cardiac events (*p* = 0.894 for any cardiac event; *p* = 0.099 for hard cardiac events).

It is widely accepted that MPI, combined with the semi-quantitative assessment of myocardial perfusion, has an important role in the prognostic evaluation of patients with known or suspected CAD. Notably, the incremental predictive value of MPI is significant even after stratifying for pre-imaging data, such as gender, age, and clinical features [18]. Previous studies have shown that the extent and severity of the inducible perfusion abnormalities represent crucial predictive variables regarding the likelihood of future cardiac events. SSS is more strongly related to mortality, while SDS was reported to be a better predictor of subsequent coronary revascularization [19]. Finally, the prognostic ability of semi-quantitative MPI parameters for major adverse cardiac, cerebrovascular, and renal events has been investigated in special patient groups, such as patients with chronic kidney disease and the elderly [20,21,22].

Doğan et al. revealed a strong correlation between semi-quantitative visual analysis of MPI findings and ICA data [23]. The authors noted that MPI quantitative analysis represents a valid tool for the detection of significant coronary stenoses in patients with stable angina. However, the researchers did not perform a prognostic analysis of their data. Moreover, Momose et al. reported that automated measurements based on quantitative gated myocardial SPECT (QGS), particularly combined with gated functional data, have incremental prognostic value, in addition to angiographic findings, in subjects referred for ICA [24]. Based on their findings, Cox univariate analysis revealed that SSS and SDS were significant predictors, while multivariate analysis indicated SDS as a significant prognostic factor (*p* < 0.001). Interestingly, MPI’s prognostic ability has been evaluated in patients who underwent percutaneous coronary intervention (PCI) or coronary artery bypass grafting (CABG). After PCI, a normal MPI study has an excellent negative predictive value for cardiac adverse events [18]. Zhang et al. noted that patients with reversible defects were at higher post-PCI risk for future events, independent of chest pain symptoms, while symptomatic patients with irreversible defects showed a higher risk for repeat revascularization [25]. Multivariate Cox analysis revealed that SSS was the best independent predictor for hard cardiac events (*p* < 0.001). Further, after CABG, MPI can detect silent progression of prognostically important disease, even in asymptomatic patients [18].

In the present study, we aimed to compare prognostic performance between the expert scoring of MPI images, combined with the automated measurements of SSS, SRS, and SDS, and the angiographic score. To the best of our knowledge, this is the first study to investigate the prognostic ability of MPI expert reading and automated analyses compared to angiographic findings. Based on our results, we demonstrated the higher prognostic ability of the combination of expert reading and automated analyses, while the angiographic score did not show a significant incremental predictive value over the above combination, at least not for the cardiac events under investigation. Therefore, prognostic information obtained through the combination of expert scoring and automated measurements may lead to better therapeutic management of patients with known or suspected CAD, possibly removing the need for invasive techniques, such as coronary angiography.

However, a limitation of our study is the potential selection bias. The study population consisted of patients clinically referred to our laboratory for the performance of MPI who also underwent coronary angiography within a 3-month period. Obviously, this patient group represents a higher-risk subset of patients with known/suspected CAD. However, the clinical value of accurate patient prognostication is even more important in patients with a higher risk of cardiac events, considering that the incremental significance of the related evidence can influence the choice of performing an invasive procedure or following medical therapy. Other potential limitations of our study include the single-center sample, the absence of external validation, and the lack of CT-based attenuation correction of acquired scintigrams.

## 5. Conclusions

Prognostication has a crucial role in the management of patients with known or suspected CAD. After comparing the prognostic ability of MPI expert scoring of SSS, SRS, and SDS (combined with automated measurements of the corresponding parameters) and the angiographic score, we demonstrated the superiority of MPI for patients’ prognostication. Moreover, there was no significant incremental predictive value of the angiographic score over the combination of expert reading and automated analyses. Therefore, our results support the pivotal contribution of MPI-derived data for CAD prognosis, with obvious effects on patients’ clinical management and therapy.

## Figures and Tables

**Figure 1 jimaging-12-00006-f001:**
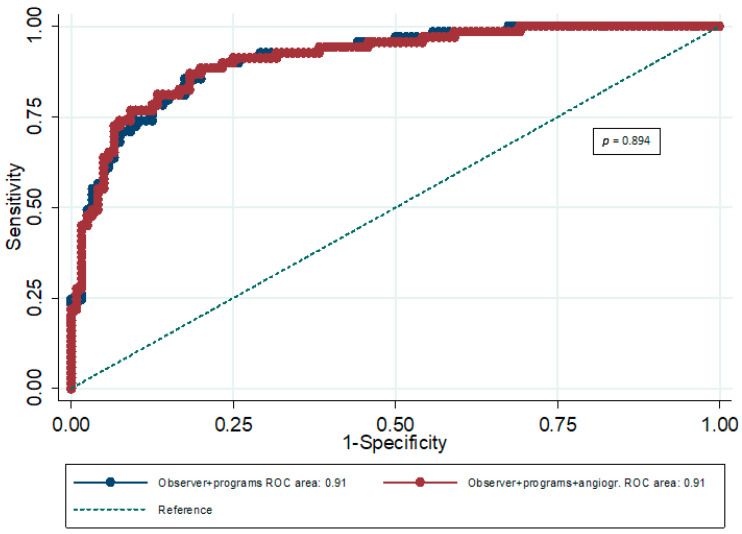
ROC curves for the combination of expert scoring (observer) and software packages (programs) and the combination of expert scoring, software packages, and angiographic score for any cardiac event.

**Figure 2 jimaging-12-00006-f002:**
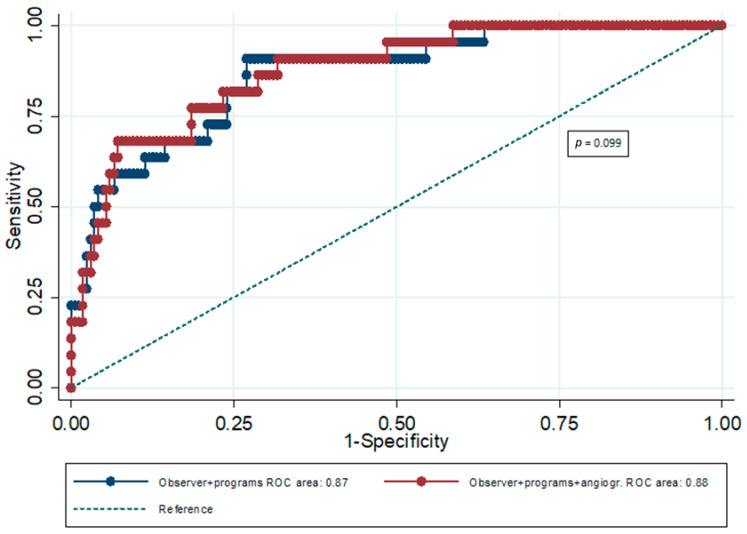
ROC curves for the combination of expert scoring (observer) and software packages (programs) and the combination of expert scoring, software packages, and angiographic score for hard cardiac events.

**Table 1 jimaging-12-00006-t001:** Hard and soft cardiac events under investigation.

Hard Cardiac Events	Soft Cardiac Events
All-cause death Cardiovascular death Non-fatal MI	PCI or CABG (≥3 months after myocardial SPECT Hospitalizations (due to unstable angina, HF, or resuscitated CA) Stroke

CA: cardiac arrest; CABG: coronary artery bypass grafting; HF: heart failure; MI: myocardial infarction; PCI: percutaneous coronary intervention.

**Table 2 jimaging-12-00006-t002:** Sample characteristics.

	N (%)
Gender	
Females	144 (38.1)
Males	234 (61.9)
Age, mean (SD)	63.8 (9.6)
BMI, mean (SD)	29.5 (5.4)
Symptoms	272 (72)
Angina	94 (24.9)
Angina-like symptoms	104 (27.5)
Dyspnea	86 (22.8)
Palpitations	74 (19.6)
Fatigue	72 (19)
Number of risk factors, median (IQR)	3 (2–4)
Smoking	148 (39.2)
Hypertension	282 (74.6)
Diabetes mellitus	130 (34.4)
Lipid disorders	300 (79.4)
Obesity	158 (41.8)
Family history of coronary artery disease	152 (40.2)
Comorbidities	86 (22.8)
Peripheral angiopathy	22 (5.8)
Stroke	28 (7.4)
Chronic obstructive pulmonary disease	50 (13.2)
Left ventricular ejection fraction, mean (SD)	0.58 (0.05)
Coronary angiography	378 (100)
Left main artery	0 (0)
Left anterior descending artery	128 (33.9)
Left circumflex artery	86 (22.8)
Right coronary artery	128 (33.9)
Angiographic Score, median (IQR)	1 (0–2)
Cardioactive agents	272 (72)
Bruce protocol	154 (44)
Pharmacologic stress	196 (56)
Hard events	44 (11.6)
All-cause death	24 (6.3)
Cardiovascular death	14 (3.7)
Non-fatal myocardial infarction (post-MPI)	18 (4.8)
Soft events	120 (31.7)
Stroke (post-MPI)	20 (5.3)
Hospitalization due to cardiac disorder (post-MPI)	104 (27.5)
Percutaneous transluminal coronary angioplasty (post-MPI)	46 (12.2)
Coronary artery bypass grafting (post-MPI)	8 (2.1)
Any cardiac event	138 (36.5)

MPI: myocardial perfusion imaging.

**Table 3 jimaging-12-00006-t003:** ROC analysis results for any cardiac event and hard events.

	Method	Index	AUC	95% CI	*p*	Optimal Cut-Off	Sensitivity (%)	Specificity (%)
Any cardiac event	ECTb	SSS	0.59	0.53–0.65	0.003	11.5	53.6	65.8
		SRS	0.54	0.48–0.60	0.241	-	-	-
		SDS	0.60	0.55–0.66	0.001	5.5	63.8	58.3
	MYO	SSS	0.67	0.61–0.73	<0.001	10.5	68.1	63.3
		SRS	0.65	0.59–0.71	<0.001	4.5	69.6	53.3
		SDS	0.60	0.54–0.66	0.001	4.5	62.3	54.2
	QPS	SSS	0.65	0.6–0.71	<0.001	6.5	66.7	59.2
		SRS	0.56	0.5–0.62	0.063	-	-	-
		SDS	0.66	0.6–0.71	<0.001	2.5	75.4	54.2
	Expert score	SSS	0.88	0.84–0.91	<0.001	4.5	89.9	75.8
		SRS	0.72	0.67–0.77	<0.001	1.5	60.9	75.8
		SDS	0.87	0.83–0.91	<0.001	4.5	84.1	79.2
	Expert score–all indexes combined		0.88	0.84–0.91	<0.001			
	Angiographic score		0.71	0.65–0.76	<0.001	0.5	76.8	52.5
Hard events	ECTb	SSS	0.63	0.54–0.73	0.004	11.5	68.2	62.3
		SRS	0.51	0.42–0.61	0.780	-	-	-
		SDS	0.67	0.59–0.76	<0.001	5.5	72.7	53.3
	MYO	SSS	0.69	0.61–0.78	<0.001	10.5	77.3	55.7
		SRS	0.67	0.59–0.74	<0.001	5.5	68.2	58.1
		SDS	0.65	0.56–0.74	0.001	6.5	54.5	69.5
	QPS	SSS	0.66	0.59–0.74	<0.001	7.5	77.3	59.3
		SRS	0.54	0.45–0.62	0.447	-	-	-
		SDS	0.67	0.59–0.75	<0.001	2.5	81.8	46.7
	Expert score	SSS	0.81	0.74–0.88	<0.001	6.5	86.4	69.5
		SRS	0.68	0.59–0.76	<0.001	1.5	63.6	65.9
		SDS	0.82	0.76–0.88	<0.001	4.5	90.9	62.3
	Expert score—all indexes combined		0.82	0.75–0.89	<0.001			
	Angiographic score		0.65	0.57–0.73	0.001	0.5	81.8	44.9

ECTb: Emory Cardiac Toolbox; MYO: Myovation; QPS: Quantitative Perfusion SPECT; SDS: Summed Difference Score; SRS: Summed Rest Score; SSS: Summed Stress Score.

**Table 4 jimaging-12-00006-t004:** ROC analysis results for any cardiac event regarding the incremental predictive ability of angiographic score.

	AUC	95% CI	*p*	P for Comparison Between AUCs
ES plus 3 software packages combined	0.91	0.88–0.94	<0.001	0.894
ES plus 3 software packages combined plus AS	0.91	0.88–0.94	<0.001	

AS: angiographic score; ES: expert score.

**Table 5 jimaging-12-00006-t005:** ROC analysis results for hard cardiac events regarding the incremental predictive ability of angiographic score.

	AUC	95% CI	*p*	*p* for Comparison Between AUCs
ES plus 3 software packages combined	0.87	0.81–0.92	<0.001	0.099
ES plus 3 software packages combined plus AS	0.88	0.82–0.93	<0.001	

AS: angiographic score; ES: expert score.

## Data Availability

The data presented in this study are available on request from the corresponding author due to privacy and ethical restrictions.

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
