# Peer review of "Long-Term Prognostic Value in Nuclear Cardiology: Expert Scoring Combined with Automated Measurements vs. Angiographic Score"

_2313-433X, 2025, doi:10.3390/jimaging12010006_

Round 1

Reviewer 1 Report

Comments and Suggestions for Authors
  • What is the role of (SSS),  (SRS), and (SDS)?
  • Abstract look very length. need not mention all the details?
  • Any justification to myocardial infarction (MI) and cardiac death is linearly?
  • Author please check the Line number 58-71.
  • Inference of Table 1. Study exclusion criteria.?
  • Need for Stress testing procedures?
  • In 17 LV segments using a 5-point scoring system Why 5-point scoring system used?
  • User mention about not used MYO software package, lease mention not used data from MYO software package?
  • In  section 2.6. Statistical Analysis add table graph to understand analysis easily.
  • Use of table 3?
  • In Discussion section need to improve provide analytic result rather than theorytical  study.

Reviewer 2 Report

Comments and Suggestions for Authors
  • Good literature review but doesn't adequately set up why this comparison (expert + automated vs angiography) is important. What is the clinical decision that would change based on these results?
  • The temporal relationship between MPI and angiography is insufficiently described and potentially problematic. The authors states patients underwent angiography "prior to or after SPECT MPI (within a 3-month period)" but doesn't specify the distribution. I suggest providing explicit data on percentage with angiography before vs. after MPI, and median time interval.
  • Logistic regression models were used to derive linear predictors - were these multivariable models? What covariates were included? How were the "combinations" of expert score + software created? Simple addition?
  • All patients underwent both MPI and angiography, creating severe selection bias, since patients referred for angiography represent a high-risk subset. This fundamentally undermines the claimed "prognostic superiority" of MPI over angiography. Acknowledge this as a major limitation. The study compares prognostic accuracy in a select population already deemed to require angiography, not in the general population of patients with known/suspected CAD
  • Event definitions are problematic. Hospitalizations (due to unstable angina, HF, or resuscitated CA) - these are vastly different endpoints with different relationships to ischemia. Moreoverm stroke is included as a "soft cardiac event" but cerebrovascular disease has different pathophysiology. Clarify event definitions and provide proper hierarchical event counts. Consider reporting time-to-first-event analyses
  • Doğan et al. revealed strong correlation between semi-quantitative visual analysis of MPI findings and ICA data" - but the present study shows MPI is better than ICA for prognosis. The discussion doesn't adequately address this apparent contradiction

Reviewer 3 Report

Comments and Suggestions for Authors
  1. Please clarify whether the visual MPI readers were fully blinded to angiographic findings and clinical follow-up information, as this directly affects validity.
  2. The rationale for including both expert scoring and automated measurements is sound, but the statistical approach for combining these inputs should be described in more detail.
  3. Consider expanding the limitations section to address the single-center sample, lack of attenuation correction, potential selection bias, and absence of external validation.
  4. The introduction could better highlight the current gaps in prognostic stratification and explain how this study fills those gaps.

Round 2

Reviewer 2 Report

Comments and Suggestions for Authors

Well done.